# Chemical Identification of Secondary Metabolites from Rhizospheric Actinomycetes Using LC-MS Analysis: In Silico Antifungal Evaluation and Growth-Promoting Effects

**DOI:** 10.3390/plants12091869

**Published:** 2023-05-02

**Authors:** Hazem S. Elshafie, Laura De Martino, Carmen Formisano, Lucia Caputo, Vincenzo De Feo, Ippolito Camele

**Affiliations:** 1School of Agricultural, Forestry, Food and Environmental Sciences, University of Basilicata, 85100 Potenza, Italy; hazem.elshafie@unibas.it; 2Department of Pharmacy, University of Salerno, Via Giovanni Paolo II 132, 84084 Fisciano, Italy; ldemartino@unisa.it (L.D.M.); lcaputo@unisa.it (L.C.); defeo@unisa.it (V.D.F.); 3Department of Pharmacy, School of Medicine and Surgery, University of Naples Federico II, Via Montesano 49, 80131 Naples, Italy; carmen.formisano2@unina.it

**Keywords:** antimicrobial activity, natural products, microbial metabolites, plant diseases, biological control

## Abstract

The rhizosphere is a rich source of actinomycetes which can produce several potential biologically active secondary metabolites. The principal goal for this research is to extract, purify, and characterize the bioactive secondary metabolites produced by three different strains of actinomycetes isolated from the rhizosphere of rosemary, black locust, and olive. The plant growth-promoting effect (PGPE) of the studied strains of actinomycetes on *Ocimum basilicum* L. (basil) and the disease-control effect on necrotic stem lesions of “black leg” caused by *Fusarium tabacinum* on basil were evaluated in silico. The cell-free culture filtrates from the studied actinomycetes isolates were evaluated in vitro for their antimicrobial activity against some common phytopathogens. The secondary metabolites obtained from the cell-free culture filtrates have been chemically characterized using high-resolution electrospray ionization of liquid-chromatography/mass-spectrometric detection (ESI-(HR)Orbitrap-MS). Results of the in silico trial showed that all studied isolates demonstrated PGPE on basil seedlings, improved some eco-physiological characteristics, and reduced the disease incidence of *F. tabacinum*. The extracted metabolites from the studied actinomycetes demonstrated antimicrobial activity in a Petri-plates assay. The chemical analysis revealed the presence of 20 different components. This research emphasizes how valuable the examined isolates are for producing bioactive compounds, indicating their putative antimicrobial activity and their potential employment as fungal biocontrol agents. In particular, the obtained results revealed the possibility of green synthesis of some important secondary metabolites, such as *N*-Acetyl-l-histidinol, Rhizocticin A, and Eponemycin, from actinomycetes. The bioactive metabolites may be successively used to develop novel bio-formulations for both crop protection and/or PGPE.

## 1. Introduction

Bioactive substances are abundantly produced by soil microorganisms [1]. Due to bacteria’s capacity to produce a variety of useful products, such as antibiotics, fungicides, herbicides, hydrolytic enzymes, antitumor, antivirals, and immune suppressants, there is increased interest in employing them for medical and agricultural applications [2,3,4]. Recently, pathogen resistance demands the discovery of novel antimicrobial substances that are effective against serious phytopathogens; thus, there is significant interest in screening new microbes with antimicrobial activity from various environments in order to explore novel potential medications against infections that are resistant to drugs. Among these new isolates are actinomycetes, unicellular filamentous Gram-positive bacteria, which are found throughout nature in a wide range of environments. Actinomycetes are prominent and significant producers of numerous biological by-products such as antibiotics and plant growth-promoting substances [5]. Actinomycetes are very similar to fungi due to their ability to form a mycelium (hyphae), but their hyphae are much smaller than fungal ones [5,6].

Actinobacteria is regarded as one of the major groups of actinomycetes, and it includes the genus *Streptomyces,* which produces several known antibiotics [4]. According to Berdy [7], the majority of the discovered bioactive substances are produced by the genus *Streptomyces*. Girão et al. [8] reported the presence of several bioactive compounds, which account for around 45% of all known microbial bioactive metabolites, and they have been isolated from actinobacteria from terrestrial sources. Girão et al. [8] investigated the ability of several actinobacteria isolated from *Laminaria ochroleucae* to control *Candida albicans* and *Staphylococcus aureus*. As a result, new actinomycetes strains can be isolated for the purpose of discovering novel bioactive substances for use in agriculture, especially for the suppression of harmful phytopathogens.

*Ocimum basilicum* L. (basil) is a culinary herb of the *Lamiaceae* family. It is native of tropical zones of central Africa and southeast Asia, and it can be also found in Mediterranean region [9]. Basil can be infected by several phytopathogens, such as *Fusarium* sp. (wilt disease), *Pythium* sp. (damping off), *Botrytis* sp. (gray mold), *Colletotrichum* sp. (black spot), and *Peronospora* sp. (downy mildew) [10]. In particular, *F. tabacinum* W. Gams (Beyma) causes necrotic stem lesions (“black leg disease”) on basil [11].

The primary goals of the present research were: (i) the extraction and the purification of the extracts produced from three studied actinomycetes isolates; (ii) the chemical characterization of the metabolites of the three isolates using HR-LC-ESI-Orbitrap-MS; (iii) the in vitro evaluation of their antimicrobial activity against some phytopathogenic strains; and (iv) the in silico evaluation of the promoting effect of the three actinomycetes strains on *O. basilicum*, as well as an evaluation of the control of “black leg disease” on basil caused by *F. tabacinum*.

## 2. Results

### 2.1. Morphological and Molecular Identification

The morphological examination, using an optical microscope, of the three pure cultures showed the typical mycelia-like structure of actinomycetes, which is characterized by branching and filamentous growth. In particular, the colony of isolate Act1 produced well-developed vegetative radial-hyphae ranged between 1.5 and 2.0 µm Ø, consisting of long, semi-straight filaments bearing colonies arranged vertically in whorls with irregular shapes (Figure 1A,B). The isolate Act2 showed a dense radial-hyphae ranged between 1.5 and 2.0 µm Ø and consisted of long, semi-straight filaments bearing colonies arranged vertically in whorls with irregular shapes (Figure 1C,D). The isolate Act3 showed a radial-hyphae and consisted of long, straight filaments bearing colonies with regular coccoid-shape (Figure 1E,F). The extracted DNA from the studied isolates and amplified by PCR produced amplicons with a molecular weight of about 350, as expected. The negative control did not show any amplification. The amplified DNA were sequenced at BMR Genomics (Padova, Italy), as previously reported by Elshafie and Camele [12]. BLAST software was used for comparing the obtained sequences with those available in GenBank. The analysis demonstrated strong similarity percentages with *Streptomyces* sp., *Streptomyces atratus*, and *Arthrobacter humicola*, which are present in GenBank. One sequence for each studied isolate was deposited in the NCBI GenBank with the following accession numbers ON241810, ON241816, and ON241806 for *Streptomyces* sp. (Act1), *Streptomyces atratus* (Act2), and *Arthrobacter humicola* (Act3), respectively.

### 2.2. Growth-Promoting and Disease-Control Effects

#### 2.2.1. Eco-Physiological Characteristics

Bacterization of all plants with the actinomycete isolates resulted in stimulation of basil seedling growth and showed higher values of eco-physiological characteristics compared to the negative control (non-bacterized plants) (Table 1). In particular, regarding the healthy seedlings, only plants inoculated with Act1 showed the highest significant values of NL and TDwS (Table 1). Whereas, plants inoculated with Act1 and Act2 showed higher significant values regarding NT, SL and TFwS (Table 1).

On the other hand, Table 1 reports also the eco-physiological characteristics of bacterized basil infected with *F. tabacinum.* In particular, infected seedlings inoculated with Act2 demonstrated higher significant values of SL, whereas infected seedlings inoculated with Act1 showed higher significant values of NT and SL (Table 1). Generally, all inoculated seedlings after infection showed moderate values regarding all measured eco-physiological parameters which considered as promising (Table 1).

#### 2.2.2. Disease-Control Effect

The plants bacterized with *Streptomyces* sp. and *A. humicola* exhibited no foliar or radical symptoms due to *F. tabacinum* infection. In fact, the disease indexes of *Streptomyces* sp. and *A. humicola* were 0.3 and 0.2%, respectively, whereas the control effects were 99.2 and 99.5%, respectively (Table 2). On the other hand, *S. atratus*-bacterized seedlings had a moderate disease index higher than 7% and a control effect higher than 79% (Figure 2 and Figure 3).

The results also showed that the C+ve (plants only inoculated with *F. tabacinum*) developed leaf chlorosis at 20 DPI, and it later turned necrotic. In addition, high percentages of leaf wilt and root necrosis were also observed at 35 DPI. In particular, a significant high percentage of symptomatic leaves was observed in the case of seedlings infected with *F. tabacinum,* where the disease index was higher than 36% compared to C-ve (plants not inoculated with *F. tabacinum* or actinomycetes isolates) and plants bacterized with actinomycetes isolates (Figure 2, Table 2). *F. tabacinum* was always re-isolated from the infected-plants.

### 2.3. Antimicrobial Assay of Metabolites

#### 2.3.1. Antifungal Activity

Crude metabolite extracts obtained from actinomycete cultural media were evaluated for their antifungal activity against some food and phytopathogenic fungi following the incorporation method (Figure 4). *A. humicola* showed the highest antifungal effect against all tested fungi. *Streptomyces* sp. showed a moderate antifungal effect, which was higher than *S. atratus*, particularly against *Monilinia fructigena*, *Rhizoctonia solani*, *Fusarium oxysporum*, *Penicillium expansum*, and *P. digitatum*. On the other hand, *S. atratus* showed a promising effect against *Sclerotinia sclerotiorum* and *P. commune*.

#### 2.3.2. Antibacterial Activity

Regarding antibacterial activity, the crude metabolite extracts were evaluated against some food and plant pathogenic bacteria following the disc diffusion method (Figure 5). The highest significant antibacterial activity of *S. atratus* was observed against *P. aeruginosa* and *X. campestris*, showing the bacterial growth inhibition (BGI) of 88.9 and 76.7%, respectively. *A. humicola* showed the highest activity against *P. fluorescens*, *X. campestris*, *P. syringae*, and *P. aeruginosa*, showing BGI of 58.9, 57.8, 55.6 and 51.1%, respectively. *Streptomyces* sp. showed moderate activity against *P. aeruginosa*, *P. syringae*, and *X. campestris*, with BGI of 36.7, 27.8 and 24.0%, respectively. In addition, the lowest activity was observed in the case of *S. atratus* against *B. cereus* and *E. coli*, with BGI of 21.1 and 20.0%, respectively. On the other hand, *Streptomyces* sp. appeared to not be active against *E. coli*, *B. cereus*, and *B. megaterium*. In case of *A. humicola*, no activity was observed against *E. coli* and *B. megaterium*.

### 2.4. ESI/(HR)orbitrap/MS Metabolic Profiles

The chemical analysis of 3 studied actinomycetes isolates revealed the presence of 20 components, as illustrated in Table 3, (Appendix A). Particularly, compound **1**: N-Acetyl-L-histidinol, identified both in *Streptomyces* sp. (Act1) and *S. atratus* (Act2) extracts, which eluted at a retention time of 0.87 min, showed a peak [M + H]^+^ ion at *m*/*z* 184.1076, corresponding to a protonated molecular ion. From the spectrum, a produced ion at *m*/*z* 166, corresponding to a loss of oxydrile group, was also evident. We noted that the compound generating the protonated ion at 184.1076 exhibited a prominent peak in all of the extracts, and this compound was previously isolated from *Streptomyces coelicolor* [13].

Compound **2**: *N*,*N*′-Diacetyl-2-deoxystreptamine was identified only in *A. humicola* (Act3) extract, and it has molecular ion [M + H]^+^ at *m*/*z* 247.1283. 2-Deoxystreptamine antibiotics constitute a large group of aminoglycoside antibiotics to which over 20 members have been assigned to date, including the paromomycins, the kanamycins, the gentamicins, the nebramycins, ribostamycin, the lividomycins, validomycin, and ambutyrosin [14].

Compound **3**: Indolactam V displayed a sodiated ion of 324.1659, which is consistent with Indolacatm V; it was only identified in *Streptomyces* sp. (Act1) extract, and it had a retention time of 1.18 min. This compound was previously isolated from *Streptomyces blastmyceticus* and showed antagonistic activity toward protein kinase C [15].

Compound **4**: Hexahydro-2*H*-pyrido [1,2-*a*]pyrazin-3(4H)-one showed a protonated molecular ion [M + H]^+^ (*m*/*z* 155.117), detected only in *S. atratus* (Act2) extract. This compound was previously found in a *Streptomyces* extract, as reported by Nithya et al. [16].

Compounds **5**,**6**: Compound **5** corresponds to valyldetoxinine, with a molecular ion [M + H]^+^ at *m*/*z* 275.159, whereas compound 6 corresponds to paromomycin, with a potassium molecular ion [M+K]^+^ at *m*/*z* 654.265. These two compounds were found only in *A. humicola* (Act3) extract. Valyldetoxinine is a member of detoxin complex, a group of depsipeptide metabolites produced by *Streptomyces caesputosus* var. *detoxus,* which is present in the soil [17]. The detoxins exhibit antimicrobial activity against some microorganisms. Paromomycin is a naturally occurring aminoglycoside antibiotic, produced by *Streptomyces rimosus* sp. *paromomycinus*, that affects both prokaryotes and eukaryotes through the chelation with A-site of ribosome [18,19]. Valyldetoxinine and paromomycin had not been previously reported in *A. humicola*.

Compound **7**: *N*-[3-[5-(2-methylpropyl)-3,6-dioxo-2-piperazinyl]propyl]-*N*′-nitro-,(2S-cis)-Guanidine was present in both extracts of *Streptomyces* sp. (Act1) and *S. atratus* (Act2), and it was eluted at 7.28 min, with a hydrate molecular ion [M + H_2_O]^+^ at *m*/*z* 332.1812 [20]. This compound was a cyclic dipeptide (CDP), widely biosynthesized during the cyclo-dipeptide synthesis cycle within prokaryotic and/or eukaryotic cells [21]. This compound was also found in *Streptomyces* strains [22].

Compound **8**: *n*-Hexyl-β-d-glucoside was detected in *Streptomyces* sp. (Act1) and *S. atratus* (Act2) extracts, with an [M + H]^+^ ion at *m*/*z* 265.1654; it is a substrate of β-glucosidases, which are hydrolytic enzymes normally present in *Streptomyces* strains that cleave β-glycosidic bonds of carbohydrates [23].

Compounds **9**–**11**: Nonactic acid (9) and homononactinic acid (11) were identified only in *S. atratus* (R3) extract, with an [M + H]^+^ ion at *m*/*z* 203.1274 and *m*/*z* 217.1430, corresponding to protonated molecular ions, respectively. These two compounds were isolated from *Streptomyces* and assayed against a panel of cancer cell lines, as reported by Lu et al. [24].

Compound **10**: (11aS)-1,2,3,11a-Tetrahydro-8-hydroxy-7-methoxy-5H-pyrrolo [2,1-c][1,4]benzodiazepin-5-one, also known as antibiotic DC81, was present in Streptomyces sp. (Act1) and S. atratus (Act2), with *m*/*z* 247.1073, and it was previously reported as an antitumor and antibiotic compound that originated from some actinomycetes [25].

Compounds **12**–**15**: These compounds were identified only in *Streptomyces* sp. (Act1) extract. Compound **12** gave an [M + H]^+^ ion at *m*/*z* 307.1646, attributed to phthoxazolins B, C, or D. These compounds were isolated from the culture broth of *Streptomyces* sp., as reported by Shiomi et al. [26], but the limitation of LC-MS is that metabolites are structural isomers which cannot be distinguished. Compound **13** was identified as Chandrananimycin D, based on an accurate mass at *m*/*z* 301.0811, and it was attributed to a protonated molecular ion [M + H]^+^: The phenoxazinone chandrananimycin D was first characterized and isolated from *Streptomyces griseus*; its antiproliferative activity has also been reported [27]. Compound **14** gave an [M + H]^+^ ion at *m*/*z* 299.0659, which was attributed to a protonated molecular ion of carboxyexfoliazone. This compound was first isolated from a wild-type *Streptomyces* strain, as reported by Abdelfattah et al. [28]. Compound **15** eluted at 10.52 min, and it had a molecular ion [M + H]^+^ at *m*/*z* 283.0705, corresponding to a protonated molecular ion of phencomycin, which was previously reported by Chatterjee et al. [29].

Compounds **16**–**18**: These compounds were detected only in *A. humicola* (Act3) extract, giving an [M + H]^+^ ion at *m*/*z* 288.2889 and an [M + H]^+^ ion at 346.2220, respectively. Particularly, compound 16 was attributed to a protonated molecular ion of 2-amino-3-hydroxyhexadecanoic acid, and has already been reported in the *Arthrobacter* genus [30]. Compound **18** was attributed to a protonated molecular ion of Maoxianamides A or B. These two compounds were isolated from *Streptomyces maoxianensis*, as reported by Li et al. [31]. Compound **17**: 1,1-Dimethylethyl 2-[2-(ethoxycarbonyl)-1-cyclopenten-1-yl]diazenecarboxylate, found only in *S. atratus* (Act2) extract, gave an [M + H]^+^ ion at *m*/*z* 269.14923.

Compound **19**: This compound was attributed to a protonated molecular ion Rhizocticin A, and it was identified both in *Streptomyces* sp. (Act1) and *S. atratus* (Act2) extracts, and it gave an [M + H]^+^ ion at *m*/*z* 352.30490. This compound is a natural phosphonate antibiotic produced by the bacterial strain *Bacillus subtilis* [32], and it is very similar to plumbemycin, isolated from *Streptomyces plumbeus,* by the amino acid (Z)-2-amino-5-phosphono-3-pentenoic acid, which is present in both antibiotics [33].

Compound **20**: This compound was attributed to a protonated molecular ion of Eponemycin, and it was identified in all three studied isolates. It gave an [M + H]^+^ ion at *m*/*z* 399.24997. This compound was reported to be produced by *Streptomyces hygroscopicus* and showed potent growth inhibition of various tumor cells [34]. Recently, Fitri et al. [35] reported that this compound is a potential candidate for a new antimalarial drug due to its efficacy against *Plasmodium berghei*.

## 3. Discussion

The obtained results are promising for biocontrol of *F. tabacinum*; the treatments with actinomycetes isolates showed a high reduction of disease symptoms on *O. basilicum* seedlings, demonstrating the control effect exerted by *Streptomyces* sp. and *Arthrobacter humicola* against *F. tabacinum*. Furthermore, the treatment with the three studied isolates of actinomycetes may induce also the resistance effect against *F. tabacinum*. In addition, the obtained results of the current study are in agreement with the results reported by Elshafie and Camele [12], where the same studied strains showed the capacity to stimulate the development of *Solanum lycopersicum* and reduce the disease incidence of *S. sclerotiorum*. The results agree with several studies which reported that many actinomycetes from the soil can inhibit some harmful phytopathogenic fungi [36,37,38].

The PGPE of the studied isolates also showed significant influences on the eco-physiological characteristics of basil plants, which may be due to synthesis of phytohormones such as gibberellic acid, Indole 3-acetic acid, and zeatine (Z) produced by symbiotic and saprophytic actinomycetes [39].

Chaudhary et al. [1] investigated the antagonistic behavior of some actinomycetes isolates obtained from various niche environments in India, and observed the bioactivity of some of the studied isolates against *Bacillus cereus*, *Shigella dysenteriae*, and Methicillin-resistant *Staphylococcus*. The same authors also noted that none of the studied isolates were able to stop mycelial growth inside cells; however, all were able to stop the extracellular growth of the filaments of the studied bacteria [1].

Regarding the secondary metabolites produced by various *Streptomyces* species, Odumosu et al. [40] reported the presence of some bioactive substances with antibiotic properties; thus, they are a potential source of novel antibiotics.

Javoreková et al. [41] investigated the addition of *Streptomyces sampsonii* and *S. flavovariabilis* to vermicompost, and the results revealed the strongest antagonistic action against a number of phytopathogenic fungi, including *Rhizoctonia solani*, *Alternaria tenuissima*, *Aspergillus niger*, and *Penicillium expansum*.

Our analysis showed the presence of several components, including phenoxazinones and a detoxin complex, to which the bioactivities could be attributed. In particular, chandrananimycin D was reported in the literature [42] for its antimicrobial activity. Valyldetoxinine, a member of detoxin complex, was distinguished by its remarkable detoxifying effect against the antibiotic blasticidin S, both in animal and plant cells [43]. Paromomycin, an antibiotic depsipeptide, can be used as a biocontrol agent to suppress soilborne diseases; the compound can also be used as plant protection agent, in particular against *Pectobacterium carotovorum,* the agent responsible for bacterial soft-rot-producing pectolytic enzymes that hydrolyze pectin between individual plant cells, and against *Phytophthora capsici*, the agent responsible for blight and fruit rot of peppers and other important commercial crops [44].

Jizba et al. [45] reported stimulatory effects of nonactic and homonanctinic acids on *Cucumis sativus* growth, and the substances were also considered later as pesticidal metabolites by Jizba and Skibova [46]. Li et al. [31] reported a moderate antifungal activity of maoxianamide A and B against *S. sclerotiorum*, the fungal phytopathogen responsible for white mold, which is a ubiquitous and highly destructive disease. In addition, rhizocticin A was reported for its effect on the inhibition of *Rhizoctonia solani*, the fungal pathogen that causes brown patch [47].

## 4. Materials and Methods

### 4.1. Actinomycetes Isolation

The three studied actinomycetes used in this study have been isolated from the rhizosphere of black locust (*Robinia pseudoacacia* L.) (Act1), rosemary (*Rosmarinus officinalis* L.) (Act2), and olive (*Olea europaea* L.) (Act3) at Potenza city (Basilicata region, Sothern Italy), following the membrane filter technique [48]. Briefly, soil samples were collected from the studied area and dried under laminar flow for one week. The soil samples were further held in a water bath at 50 °C for 30 min to destroy vegetative microorganisms. Starch casein media (SCM) was prepared for isolating actinomycetes as follows: [g/L]: starch (10); K_2_HPO_4_ (2); KNO_3_ (2); casein (0.3); MgSO_4_.7H_2_O (0.05); CaCO_3_ (0.02); FeSO_4_.7H_2_O (0.01); agar (15). The pH was adjusted to 7.0. The prepared media was autoclaved at 121 °C for 15 min and then was poured in plates at 45 °C and allowed to settle. The isolation on agar plates was carried out by placing filter (mixed cellulose ester, pore size = 0.2 mm, Advantec) on the center of the Petri dishes, and 0,5 g of soil was sprinkled on the membrane filters pre-moistened with 100 µL of sterile distilled water (SDW). All plates were incubated at 28 °C for 4 days, and then the membrane filters were removed. The plates were further incubated until the actinomycete colonies became visible, after which a sub-culturing was performed. The isolates were sub-cultured and conserved in peptone yeast calcium agar (PY-CA), which contained [g/L]: peptone (5), yeast extract (3), and calcium chloride (0.7). This method is preferred due to its selectivity of actinomycetes and due to the inability of other microbes to produce mycelia that can penetrate the membrane filter.

### 4.2. Morphological and Molecular Identification

The morphological identification of the three actinomycetes isolates was carried out previously based on their microscopic features using a light microscope (Axioskop—ZEISS, Oberkochen, Germany). Molecular identification based on PCR techniques was carried out as follows: Briefly, PCR primers Y1 (5′-TGG CTC AGG ACG AAC GCT GGC GGC-3′) and Y2 (5′-CCT ACT GCT GCC TCC CGT AGG AGT-3′) [49] were used to amplify a 348-bp fragment from the 16S rRNA gene. The PCR reaction was performed under the following conditions: 1 cycle of 94 °C for 5 min, followed by 34 cycles of 94 °C for 30 s; 57 °C for 30 s; and 72 °C for 1 min, with a final extension step (1 cycle) of 72 °C for 5 min. Preparation of the samples for PCR was performed as described by Ward et al. [50], and the eventual obtained amplicons were observed by electrophoresis in a 1.5% agarose gel.

### 4.3. Growth-Promoting and Disease-Control Effects

A greenhouse trial was undertaken out to evaluate the PGPE of the three isolates for basil, as well as to evaluate their disease-control effect (DC) against the necrotic stem lesions (“black leg”) caused by *F. tabacinum* (Appendix A). Basil seeds were surface sterilized with ethanol (70%), rinsed three times with sterile distilled water, and then were sowed in polystyrene seed trays. The greenhouse’s temperature and relative humidity were maintained at 24 ± 2 °C and 60–70%, respectively, during the entire experiment.

Regarding actinomycetes treatment, a suspension 10^6^ CFU.mL-1 of each isolate obtained from 5-days-fresh PY-Ca culture and inoculated into minimal mineral (MM) media was prepared as follows [g/L]: dipotassium phosphate (10.5), potassium dihydrogen phosphate (4.5), ammonium sulfate (1.5), trisodium citrate dihydrate (0.5), magnesium sulfate (0.2), and dextrose (5.0), and the pH value was adjusted to 7.0. The broth culture was incubated in a rotary incubator at 28 °C and 180 rpm for 8 days. The broth culture (100 mL/pot) was poured into the basil-rhizosphere at 15 days post-seed germination (DPSg).

For artificial fungal-infection, a conidial-suspension (10^8^ spore/mL) of *F. tabacinum* was inoculated in a potato dextrose broth (PDB) flask and incubated under stirring (180 rpm) for 7 days at 22 °C. In total, 50 mL of broth was inoculated into the basil-rhizosphere 10 days after actinomycetes treatment. Twenty seedlings were used for each experiment: (i) untreated healthy; (ii) treated only with actinomycetes; and (iii) treated only with fungi.

At the end of the trial, plant growth was examined for the eco-physiological characteristics 40 DPSg following the method explained by Elshafie et al. [51]. The stem length (SL), leaf number (NL), twigs number (NT), total shoot fresh-weight (TFwS) and total shoot dry-weight (TDwS) were measured. The disease incidence was monitored daily for 15 days post-infection (DPI) using the following scale: 0 = less than 5% symptomatic leaf; 1 = 6 to 20% of symptomatic leaf; 2 = 21 to 50% of symptomatic leaf; 3 = 51 to 80 % of symptomatic leaf; and 4 ≥ 80% of symptomatic leaf [12]. Using Formula 1, the infection proportion (IP%) was calculated, and Formulas 2 and 3 were used for evaluating the disease index (DI%) and control effect (CE%), respectively [52].
(1)$$IP\left(\%\right)=\frac{NSL}{TLN}\times 100;$$
(2)$$DI\left(\%\right)=\left[\Sigma \frac{\left(Scalevalue\times NSL\right)}{Hi.S\times TL}\right]\times 100$$
(3)$$CE\left(\%\right)=\left[\frac{Di.P-Di.B}{Di.P}\right]\times 100$$
where: NSL = number of symptomatic leaves; TLN = total leaf number; TL = total number of leaves; Hi.S = highest scale; DI-P = disease index of pathogen treatment; DI-B = disease index of actinomycetes-treated.

### 4.4. Extraction of Metabolites

The secondary metabolites of the three studied strains were obtained following the method of Lavermicocca et al. [53], with minor changes, as follows: an Erlenmeyer flask was filled with 140 mL of MM broth and seeded with 2.0 mL of each actinomycete suspension at 10^7^ CFU/mL. It was incubated in a rotary incubator at 28 °C and 180 rpm for 8 days. The broth culture of each isolate was centrifuged at 20,000× *g*/10 min, the precipitate was discarded, and the upper-phase was filtered using Millipore 0.22 μM. The purified filtrate was extracted in equal volume of suitable organic solvent (ethyl-acetate) by shaking for 5 min and separated using a separator funnel. The combined organic fractions were concentrated using an evaporator (Heidolf 2000, Schwabach, Germany) at 180 rpm/80 °C for 20 min. The dried extracts were resuspended in 1 mL of sterile distilled water [54].

### 4.5. Microbicidal Test of Metabolites

#### 4.5.1. Antifungal Assay

The in vitro antifungal activity of the extracted metabolites from the three studied isolates was evaluated against the following phytopathogenic fungi: *M. fructigena*, *R. solani*, *F. oxysporum*, *S. sclerotiorum*, *P. expansum*, *P. digitatum*, and *P. commune*, using the incorporation method [55,56]. In total, 10 μL of each extract, in concentrations of 100 and 50%, was deposited on a potato dextrose agar (PDA) [57] pre-inoculated with each tested fungal disc. To assess the antifungal activity, we determined the diameter of the grown mycelium in millimeters. The inhibition percentage of fungal growth (FGI%) was calculated following Formula (4) [58]. Cycloheximide was used as positive control (C+ve) at 100 µg/mL.
(4)$$FGI\left(\%\right)=D.MGt/D.MGc\times 100$$
where: FGI (%) is the mycelium inhibition percentage: D.MGt is the mean diameter of the fungal mycelium in the treated Petri dish (mm); and D.MGc is the mean diameter of the mycelium control Petri dish (mm).

#### 4.5.2. Antibacterial Assay

The in vitro antibacterial activity of the extracts was evaluated against the following bacteria: *Pseudomonas syringae*, *P. fluorescence*, *P. aeruginosa*, *E. coli*, *Bacillus cereus, B. megaterium*, and *Xanthomonas campestris*, using the disc diffusion method [59,60,61]. King’B (KB) nutrient media was used for reculturing the studied bacterial strains [62]. The bacterial suspensions were prepared in SDW and adjusted to 10^6^ CFU/mL. In total, 4 mL of each bacterial suspension diluted in soft agar (0.7%) at 9:1 *v*/*v* were poured into a KB Petri dish (90 mm). A total of 15 μL of each extract, at concentrations of 100 and 50%, was deposited on filter discs (6 mm-OXOID) previously placed on plates and left for 30 min under laminar flow. Finally, the bactericidal effect was evaluated by measuring the diameter of the inhibition zone (D.Iz) in millimeters compared to tetracycline (1600 µg/mL), used as C+ve. The bacterial growth inhibition percentage (BGI%) was calculated following Formula (5). All tested treatments were carried out in triplicate ± standard deviations (SDs).
BGI (%) = D.Iz/D.Cc × 100 (5)
where: BGI (%) represents the bacterial inhibition percentage; D.Iz is the mean diameter of inhibition zone in the treated Petri dish (mm); and D.Cc is the mean diameter of the bacteria grown in control Petri dish (mm).

### 4.6. ESI/(HR)orbitrap/MS Metabolic Profiles

The qualitative analysis of three studied actinomycetes extracts was performed by high performance liquid chromatography-mass spectrometry (HPLC-MS) using an LTQ-XL Ion Trap mass spectrometer (Thermo Fischer Scientific Spa, Rodano, Italy) equipped with an Ultimate 3000 HPLC (Agilent Technology, Cernusco sul Naviglio, Italy). Chromatographic separation was obtained using a Kinetex Polar C_18_ column (100 × 3.0 mm, 100 Å, 2.6 µm) (Phenomenex, Torrance, CA, USA). The injection volume was 0.5 mL/min and a mobile phase consisting of a combination of A (0.1% formic acid in water, *v*/*v*) and B (0.1% formic acid in acetonitrile MeCN); a linear gradient, which ranged between 5 and 60% B in 25 min, from 60 to 95% B in 10 min, and held at 95% B for 5 min, was used. The mass spectrometer was set in positive ion mode. ESI source conditions were the following: capillary voltage −48 V; tube lens voltage −176.47; capillary temperature 280 °C; sheath 15 and auxiliary gas flow (N_2_) 5; sweep gas 0; and spray voltage 5. MS spectra were obtained at 30,000 resolutions by full-range acquisition with a scan range between 150 and 1500 *m*/*z*.

### 4.7. Statistical Analysis

For the statistical analysis, the collected results were analyzed via a one-way ANOVA using Statistical Package for the Social Sciences (SPSS) version 13.0, 2004 (Chicago, IL, USA). The Tukey’s B post hoc multiple comparison test was applied to determine the significance level with a probability of *p ≤* 0.05.

## 5. Conclusions

This research revealed the biological activity of actinomycetes, especially *Streptomyces* and *Arthrobacter*. This study also highlighted the value of the new studied strains in terms of their capacity to produce significant bioactive by-products that can be employed as biocontrol agents against several harmful fungi, including *Fusarium* species. In conclusion, some metabolites, detected in the extracts of the studied isolates, were recognized to be effective against phytopathogenic fungi and bacteria. This may represent the potential for their use in future sustainable strategies to control outbreaks in a vast range of crops. This research demonstrated the possibility of green synthesis of some important secondary metabolites, such as N-Acetyl-L-histidinol, rhizocticin A, and Eponemycin from actinomycete. In particular, N-Acetyl-L-histidinol (detected in the current study from *Streptomyces* sp. and *S. atratus*), is considered an important derivative of the primary metabolite L-Histidinol, in agreement with Ballio et al. [13], who reported that N-acetyl-L-histidinol was produced in cultures of *Streptomyces coelicolor*. Regarding rhizocticin A (detected in the current study from *Streptomyces* sp. and *S. atratus*), it is considered a natural phosphonate antibiotic and hydrophilic phosphono-oligopeptides extracted from different biocontrol agents such as *Bacillus subtilis* [47]. Eponemycin (detected in the current study from the three studied isolates) is considered an important antibiotic with specific in vivo antitumor effects against B16 melanoma [34]. These potential bioactive compounds may be used to develop new commercial formulations, either as plant-growth promoters or for crop protection.

## Figures and Tables

**Figure 1 plants-12-01869-f001:**
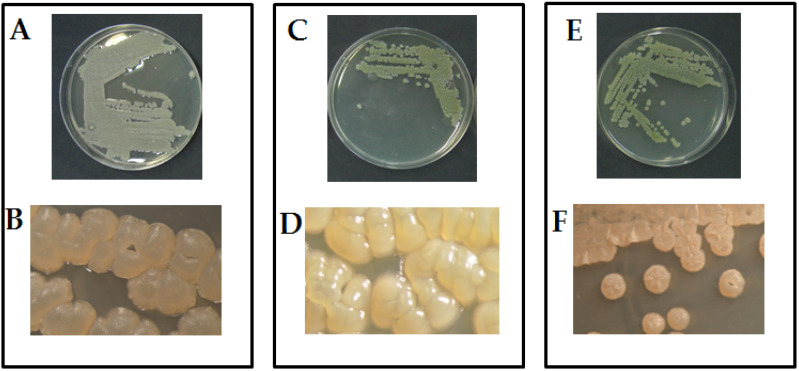
Microscopic morphological features of the three studied actinomycetes isolates. (**A**,**C**,**E**) are cultures of *Streptomyces* sp. (Act1), *Streptomyces atratus* (Act2), and *Arthrobacter humicola* (Act3) grown in PY-CA media, respectively. (**B**,**D**,**F**) are the colonies of *Streptomyces* sp. (Act1), *Streptomyces atratus* (Act2), and *Arthrobacter humicola* (Act3), respectively, examined under stereo microscope (1000×).

**Figure 2 plants-12-01869-f002:**
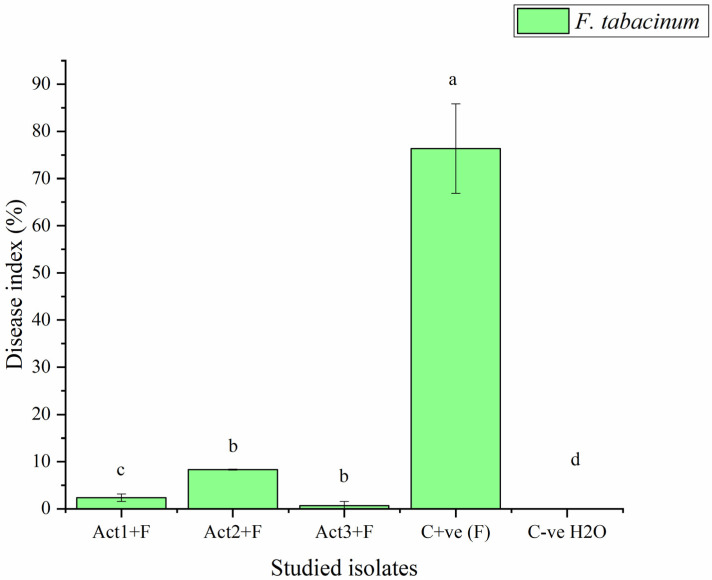
Disease index of basil infected with *F. tabacinum.* Where: Act1+F; Act2+F; Act3+F are inoculated plants with *Streptomyces* sp., *S. atratus* and *A. humicola*, respectively. Bars with different letters between different treatments are significantly different at *p* < 0.05 according to Tukey’s B multiple comparison post hoc test using SPSS software. Values are mean of 3 replicates ±SDs.

**Figure 3 plants-12-01869-f003:**
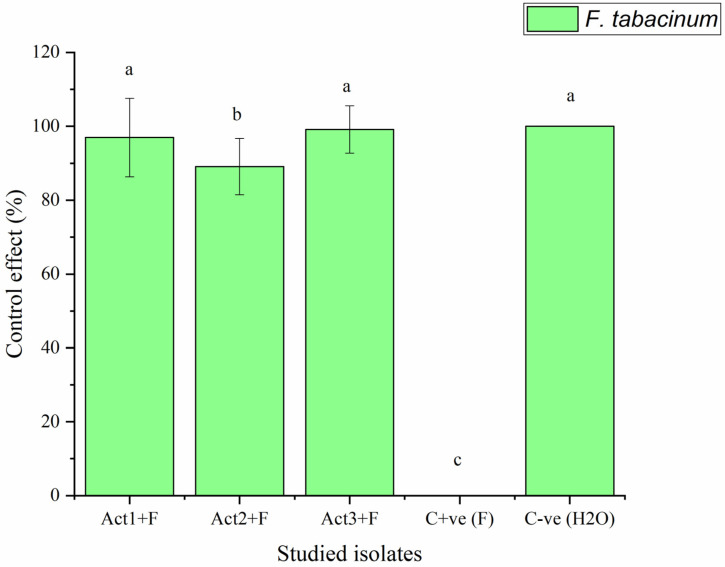
Control effect of basil after infection with pathogen. Bars with different letters between different treatments are significantly different at *p* < 0.05 according to Tukey’s B multiple comparison post hoc test using SPSS software. Values are mean of 3 replicates ±SDs.

**Figure 4 plants-12-01869-f004:**
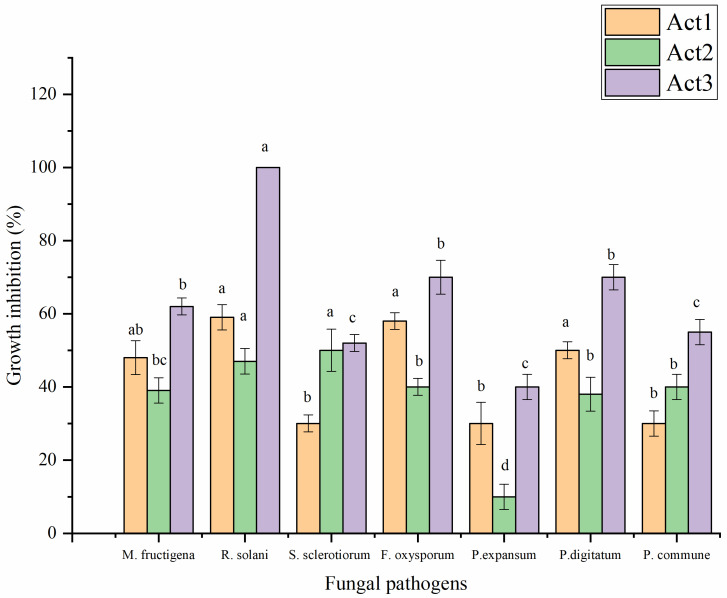
Antifungal activity of extracted metabolites. Bars with different letters between different phytopathogenic fungi for each treatment are significantly different at *p* < 0.05 according to Tukey’s B multiple comparison post hoc test using SPSS software. Values are mean of 3 replicates ±SDs.

**Figure 5 plants-12-01869-f005:**
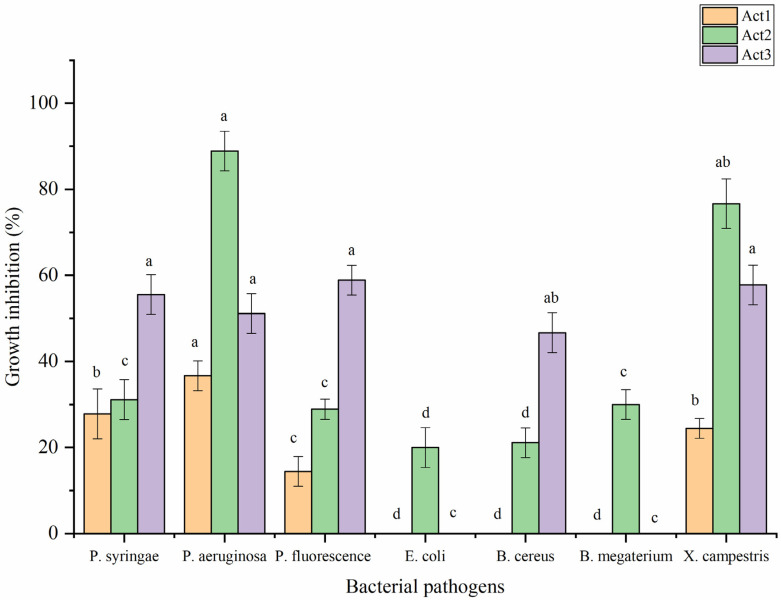
Antibacterial activity of extracted metabolites. Bars with different letters between different phytopathogenic bacteria for each treatment are significantly different at *p* < 0.05 according to Tukey’s B multiple comparison post hoc test using SPSS software. Values are mean of 3 replicates ±SDs.

**Table 1 plants-12-01869-t001:** Eco-physiological characteristics of basil after actinomycetes inoculation for healthy and infected plants.

Eco-Physiological Characteristics	Act1: *Streptomyces* sp.		Act2: *A. humicola*		Act 3: *S. atratus*		Control
	Healthy	Infected	Healthy	Infected	Healthy	Infected	
NL (n)	146 ± 7 ^a^	113 ± 12 ^b^	111 ± 10 ^b^	105 ± 5 ^b^	109 ± 9 ^b^	85 ± 7 ^c^	85 ± 9 ^c^
NT (n)	7 ± 1 ^a^	4 ± 2 ^ab^	7± 1 ^a^	5 ± 1 ^ab^	3 ± 1 ^b^	3 ± 1 ^b^	2 ± 1 ^b^
SL (cm)	45 ± 5 ^a^	39 ± 4 ^ab^	47 ± 3 ^a^	41 ± 6 ^a^	34 ± 4 ^ab^	27 ± 2 ^b^	34 ± 4 ^ab^
TFwS (g)	241 ± 13 ^a^	114± 15 ^bc^	246 ± 7 ^a^	99 ± 9 ^c^	191 ± 5 ^b^	111 ± 27 ^bc^	75 ± 4 ^c^
TDwS (g)	40 ± 4 ^a^	23 ± 2 ^b^	31 ± 4 ^ab^	18 ± 6 ^b^	20 ± 3 ^b^	14 ± 2 ^c^	15 ± 2 ^c^

NL: number of leaves; NT: twigs count; SL: stem length; TFwS: total fresh weight of shoot; TDwS: total dry weight of shoot. Values followed by different letters in each row are significantly different at *p* < 0.05 according to Tukey’s B multiple comparison post hoc test using SPSS software. Values are mean of 3 replicates ±SDs.

**Table 2 plants-12-01869-t002:** Determination of disease index and control effect on basil after different applications.

Treatments	Disease Index	Control Effect
	DI %	CE %
Act1: *Streptomyces* sp.	2.3 ± 0.8 ^c^	96.9 ± 10.6 ^a^
Act2: *A. humicola*	0.7 ± 0.1 ^d^	99.1 ± 7.6 ^a^
Act3: *S. atratus*	8.3 ± 0.9 ^b^	89.1 ± 6.4 ^ab^
Cont. +ve (F)	76.4 ± 9.5 ^a^	0.0 ± 0.0 ^c^
Cont. -ve (H_2_O)	0.0 ± 0.0 ^e^	100.0 ± 0.0 ^a^

Where: Cont. +ve (F): control positive (plants treated with fungal pathogens); Cont. -ve (H_2_O): control negative (healthy plants); DI% = disease index; CE% = control effect. Values followed by different letters in each column are significantly different at *p* < 0.05 according to Tukey’s B multiple comparison post hoc test using SPSS software. Values are mean of 3 replicates ±SDs.

**Table 3 plants-12-01869-t003:** Chromatographic analysis of extracted secondary metabolites from studied actinomycetes isolates.

No.	Retention Time (min)	Measured *m*/*z*	Molecular Formula	Identification	Act1	Act2	Act3
**1**	0.87	184.1075	C_8_H_13_N_3_O_2_	*N*-Acetyl-l-histidinol	X	X	
**2**	1.17	247.12834	C_10_H_18_N_2_O_5_	*N*,*N*′-Diacetyl-2-deoxystreptamine			X
**3**	1.18	324.16592	C_17_H_23_N_3_O	Indolactam V	X		
**4**	1.19	155.11748	C_8_H_14_N_2_O	Hexahydro-2*H*-pyrido [1,2-*a*]pyrazin-3(4*H*)-one		X	
**5**	2.30	275.15955	C_12_H_22_N_2_O_5_	Valyldetoxinine			X
**6**	7.11	654.2653	C_23_H_45_N_5_O_14_	Paromomycin			X
**7**	7.28	332.18118	C_12_H_22_N_6_O_4_	Guanidine, *N*-[3-[5-(2-methylpropyl)-3,6-dioxo-2-piperazinyl]propyl]-*N*′-nitro-, (2S-cis)	X	X	
**8**	7.73	265.16544	C_12_H_24_O_6_	*n*-Hexyl-β-D-glucoside	X	X	
**9**	7.88	203.12737	C_10_H_18_O_4_	Nonactinic acid		X	
**10**	8.47	247.10730	C_13_H_14_N_2_O_3_	(11a*S*)-1,2,3,11a-Tetrahydro-8-hydroxy-7-methoxy-5*H*-pyrrolo [2,1-*c*][1,4]benzodiazepin-5-one	X	X	
**11**	8.61	217.14296	C_11_H_20_O_4_	Homononactinic acid		X	
**12**	8.81	307.16461	C_16_H_22_N_2_O_4_	Phthoxazolins B, C and D	X		
**13**	9.43	301.08112	C_15_H_12_ N_2_O_5_	Chandrananimycin D	X		
**14**	9.75	299.06592	C_15_H_10_N_2_O_5_	Carboxyexfoliazone	X		
**15**	10.52	283.07053	C_15_H_10_O_4_N_2_	Phencomycin	X		
**16**	11.32	288.2889	C_16_H_33_NO_3_	2-Amino-3-hydroxyhexadecanoic acid			X
**17**	11.45	269.14923	C_13_H_20_N_2_O_4_	1,1-Dimethylethyl 2-[2-(ethoxycarbonyl)-1-cyclopenten-1-yl]diazenecarboxylate		X	
**18**	12.36	346.2220	C_17_H_31_O_6_N	Maoxianamide A or B			X
**19**	12.20	352.30490	C_11_H_22_N_5_O_6_P	Rhizocticin A	X	X	
**20**	14.09	399.24997	C_20_H_34_N_2_O_6_	Eponemycin	X	X	X

## Data Availability

Not applicable.

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
