# Peer review of "Chemical Identification of Secondary Metabolites from Rhizospheric Actinomycetes Using LC-MS Analysis: In Silico Antifungal Evaluation and Growth-Promoting Effects"

_plants, 2023, doi:10.3390/plants12091869_

Round 1
Reviewer 1 Report
The presented study is aimed at a search of bioactive metabolits directed against phytopathogenic fungi and bacteria. To find such bioactive metabolites, extracts of growth media after cultivation of three rhizosphere isolates of actinomycetes: Streptomyces sp., Streptomyces atratus and Arthrobacter humicola, were subjected to chemical analysis. In these extracts, the authors managed to detect and chemically isolate a set of useful untifungal and antibacterial antibiotics against phytopathogenic fungi and bacteria, which can supress several harmfull fungi, including Fusarium species. The study is of interest to biotechnologists and microbiologists of the food industry. The language is very poor, and it is difficult to understand the content in many places. In some places whole chunks of sentences are missing. A huge number of corrections need to be made, which are listed below. Note the following designations: <...> - for inclusion and ]hgdvgv[ - for deletion:
Introduction
53: ]know[ <known> antibiotics
56: <and> have been
Results
82: and ]consists[ <consisted>
83-84: <and> ]consists[ <consisted>
85: ]The sequences of the[ <The> amplified DNA ]have been carried out in[ <were sequenced at>
86: software ]has been[ <was> used
101-104: A mistakenly formed sentence. I corrected this sentence as follows:
Bacterization of all plants with the actinomycete isolates resulted in stimulation of basil seedlings growth ...
114-123: I strongly recommend to merge Table1 and Table 2 results in a single Table 1. The column with headlines "Eco-Physiological Characteristics" should be relocated to the left side and oriented vertically. The column "Actinomycetes isolates" should be relocated to the right side and oriented horizontally. And below this line under each Act strain should be two windows for healthy plant and infected plant.
117; 121-122; 143; 148-149; 154; 182; 187: Please include a specific explanation of what the mysterious phrase "Values followed by different letters in each vertical column are significantly different at p < 0.05 according to SPSS software" means. It is unclear what, in particular, each letter "a,b,c" means.
126: ]A. humicola[ should be changed to italicized font <A. humicola>
142: ]treated[ plants <treated with> fungal pathogens
147: with <pathogen> F. tabacinum
148: ]A[ S. atratus and < A.> humicola
160: What were these extracts? Please explain. Is it correct? ]The[ <Crude metabolite> extracts <obtained from actinomycete cultural media>
167: the <crude metabolite> extracts
169: significant <antibacterial> activity
170: Dear authors, there is a general principle to present a deciphering of all abbreviations when they first appear in the text. The abbreviation BGI has many meanings, for example, "BGI sequencing platform", "Baltic Group International (BGI)" and so on. You should modify the following phrase as shown below and correct numerous errors:
... showing <the bacterial growth inhibition (BGI)> ]BGI% at[ by 88.9 and ]at[ 76.7%,
171-172; 173; 175: ]BGI% at[ <BGI by>
176: ]resulted[ <appeared to be>
204; 205: ]Indolacatm V[ <Indolactam V>
208: What does that mean "plue protonated ion" ? Did you mean "pure protonated ion" or something else?
214: ]was[ <is> a member
218: I have improved the phrase. Is it correct? ... both ]prokaryotic and eukaryotic[ <prokaryotes and eukaryotes> ...
224: ]was[ <is> a cyclic
225: ]synthsis[ <synthesis>
246: ]it also reported for[ its antiproliferative activity <has also been reported>
248: was firstly ]report[ <isolated>
255: already reported ]previously[
260: compound ]was[ attributed
262: <and> gave
264: ]for[ <by the> amino acid
266: ]was[ attributed
269: inhibition ]against[ <of> various
Discussion:
290: The names of organisms should not be abbreviated when they are first introduced in the text: ]S. lycopersicum[ <Solanum lycopersicum>, ]S. sclerotiorum[ <Sclerotinia sclerotiorum>
297: showed <their> significant
301: ]mycelium from growing[ <mycelial growth>
302: ] from the examined bacteria[ <of the bacteria studied>
307: ]The analysis which revealed the presence[ <Our analysis showed the presence >
308: ]therefore a potential[ <so they are a potential>
311: ]in charge of[ <responsible for>
313-314: Part of the phrase is absent: to which the bioactivities could have ???
319; 320-321: ]the[ responsible
Methods
340: methods ]-[ based <on> PCR techniques
342: (PY-CA) <add dot> ]nutrient[ <Nutrient> media ]contains[ <contained>
348: were ]adjusted[ <maintained at>
355: ]were[ <was> poured
358: ]in agitation[ <under stirring>
361: ]have been[ <were> used
363: ]going to measure[ <measuring>
383: ]and[ <was> left
387: ]and shacked[ <by shaking> for 5 min <and separated> using a separator funnel.
388: ]by[ using
395: ]Ten[ <10> μL of each extract, ]at concentration[ <in> concentrations <of> 100% and 50%
397-398: ]the diameter of mycelium grew in Millimetre[ <we determined the diameter of the grown mycelium in millimeters.>
411: ]Four[ <4> mL
412: ]Fifteen[ <15> μL
412: at ]concentration[ <concentrations>
414: ]The eventually[ <Finally, the> bactericidal effect was evaluated <by> measuring the diameter of the inhibition zone (D.Iz) in ]Millimetre[ <millimeters>
Conclusions:
441: ]serious[ <harmful> fungi
442: in ]the extracted[ extracts of
442: recognized ]as[ <to be> effective
451: <it> is considered
The same as for authors
Author Response
Dear Reviewer
Thanks so much for your accurate revision and valuable comments.
Please see the attachment.

Reviewer 2 Report
The use of actinomycetes is widespread both for crop applications in the agriculture sector and in the production of antibiotics for medical use. However, the manuscript needs major revision prior to be accepted.
Major comments:
· The introduction and discussion sections mix the use of actinomycetes for agricultural and medicinal use, although the study only analyzes their use against phytopathogenic bacteria and fungi.
· Brief materials and methods. The first part of the results uses 17 lines just to explain morphological characteristics and a sequence alignment, but not even the "Membrane Filter Technique", or the "molecular methods-based PCR techniques" (gene and primers used, conditions...) are briefly explained.
· It is difficult to follow trial 4.2 (Plant growth-promoting and disease-control effects). It is highly recommended to include a figure indicating the treatments performed and the names given throughout the article for each of them.
· Talks about Bacillus megaterium as a “pathogenic bacteria”, a well-known plant growth-promoting bacteria (PGPR), which is used commercially as biofertilizer. Same for B. cereus.
· The statistical analysis performed is not described at any point.
· The materials and methods describe that during the greenhouse trial the lesions caused by F. tabacium are evaluated (lines 344 - 346), while the results describe the effects caused by S. sclerotirum (line 127).
Minor comments:
- Lines 76 – 93, excessive for the results described.
- Homogenize font size of Figure captions (see figure 1), and the text in Figures (see F. tabacium in Figures 2 and 3).
- Homogenize the citation of figures and tables in the text (see lines 106 and 107).
- Line 126, “humicola” in italic.
- Extend the discussion section, and discuss with the agricultural sector, since all the assays have been carried out in plants or with phytopathogenic microorganisms.
- Include Tl, SL and N meaning in Formula 1 - 3 caption (lines 375 - 376).
Minor editing of English language required, such as:
- Line 16, the goal is singular: “The principal goal for this research is to extract…”.
- Review lines 397 - 398.
- Cultures are not "left in Rotatory-Incubator", are incubated at xºC, x rpm for x days/hours...
Author Response

(The authors gave the same response as above.)

Round 2
Reviewer 2 Report
Authors have properly corrected previous comments.